# A cluster feasibility trial to explore the uptake and use of e-cigarettes versus usual care offered to smokers attending homeless centres in Great Britain

Lynne Dawkins[1]*, Linda Bauld[2], Allison Ford[3], Deborah Robson[4], Peter Hajek[5], Steve Parrott[6], Catherine Best[3], Jinshuo Li[6], Allan Tyler[1], Isabelle Uny[3], Sharon Cox[1]¤

1 Centre for Addictive Behaviours Research, London South Bank University, London, England, 2 Usher Institute and SPECTRUM Consortium, Old Medical School, University of Edinburgh, Edinburgh, Scotland, 3 Institute for Social Marketing and Health, Faculty of Health Sciences and Sport, University of Stirling, Stirling, Scotland, 4 National Addiction Centre and SPECTRUM Consortium, Addictions Department & ARC South London, Institute of Psychiatry, Psychology & Neuroscience, King's College London, London, England, 5 Wolfson Institute of Preventive Medicine, Barts and The London School of Medicine and Dentistry, Queen Mary University of London, London, England, 6 Department of Health Sciences, University of York, York, England

¤ Current address: Department of Behavioural Science and Health, University College London, London, England

* dawkinl3@lsbu.ac.uk

**Data Availability Statement:** Our data are available on the LSBU institutional repository under the following DOI:10.18744/lsbu.8q255 and can be

## Abstract

Smoking rates in the UK are at an all-time low but this masks considerable inequalities; prevalence amongst adults who are homeless remains four times higher than the national average. The objective of this trial was to assess the feasibility of supplying free e-cigarette starter kits to smokers accessing homeless centres and to estimate parameters to inform a possible future larger trial. In this feasibility cluster trial, four homeless centres in Great Britain were non-randomly allocated to either a Usual Care (UC) or E-Cigarette (EC) arm. Smokers attending the centres were recruited by staff. UC arm participants (N = 32) received advice to quit and signposting to the local Stop Smoking Service. EC arm participants (N = 48) received an EC starter kit and 4-weeks supply of e-liquid. Outcome measures were recruitment and retention rates, use of ECs, smoking cessation/reduction and completion of measures required for economic evaluation. Eighty (mean age 43 years; 65% male) of the 153 eligible participants who were invited to participate, were successfully recruited (52%) within a five-month period, and 47 (59%) of these were retained at 24 weeks. The EC intervention was well received with minimal negative effects and very few unintended consequences (e.g. lost, theft, adding illicit substances). In both study arm, depression and anxiety scores declined over the duration of the study. Substance dependence scores remained constant. Assuming those with missing follow up data were smoking, CO validated sustained abstinence at 24 weeks was 3/48 (6.25%) and 0/32 (0%) respectively for the EC and UC arms. Almost all participants present at follow-up visits completed data collection for healthcare service and health-related quality of life measures. Providing an e-cigarette

found here: https://openresearch.lsbu.ac.uk/item/8q255.

**Funding:** This study is funded by the National Institute for Health Research Public Health (project reference: 17/44/29). The views expressed are those of the authors and not necessarily those of the NIHR or the Department of Health and Social Care. The funders had no role in study design, data collection and analysis, decision to publish, or preparation of the manuscript.

**Competing interests:** PH received research funding from and provided consultancy to manufacturers of stop-smoking medications. LD has provided consultancy for the pharmaceutical industry relating to the development of smoking cessation products. This does not alter our adherence to PLOS ONE policies on sharing data and materials. SC, LB, AF, DR, SP, CB, AT, JL, IU, SP have no competing interests.

starter kit to smokers experiencing homelessness was associated with reasonable recruitment and retention rates and promising evidence of effectiveness and cost-effectiveness.

## Introduction

Although smoking rates in the UK general population are on the decline, reaching a historic low of 14.7% in 2018 [1], this masks significant inequalities with stubbornly high prevalence rates among some of the most disadvantaged in society [2]. Smoking is a leading cause of health inequality and smoking-related deaths are two to three times higher among disadvantaged groups [2]. Those in the most deprived groups are also more highly nicotine dependent [3], make fewer quit attempts, and are less successful when they do make an attempt [4, 5]. Smoking prevalence amongst those experiencing homelessness has been estimated at around 57–82% based on studies predominantly deriving from the US [6]). In the UK, homeless.org estimates prevalence at 78% [7].

Despite the high smoking prevalence amongst those experiencing homelessness, interest in quitting and desire to stop is similar to the general population of smokers [8–10]. Nevertheless, there have been no smoking intervention studies specifically focusing on homeless people in the UK; the few studies that have explored the best ways to help people experiencing homelessness to quit smoking derive mainly from the US. Working with this population can be challenging for research; engagement with health services, including Stop Smoking Services (SSS) [7, 11], is generally poor and smokers experiencing homelessness may be difficult to recruit and retain in research studies. Two studies from the US and one from Australia have explored a range of interventions for smoking cessation in people experiencing homelessness including motivational interviewing, cognitive behavioural therapy, quit lines, nicotine replacement therapy (NRT) and/or other pharmacotherapies [12–14]. Point prevalence (24 hour or 7 day) abstinence rates at 6 months were low, ranging between 4 and 13.6%. A further US study in a small sample of homeless veterans reported a much higher 26 week past 7-day point prevalence abstinence rate of 45% (9/20) using contingency management (participants could earn up to $815 for CO verified abstinence) plus NRT, bupropion and a smartphone app [15]. In the only study reporting 6-month sustained abstinence, quit rates were 0% [13].

In a study of 24 low socioeconomic status smokers in Australia [16] feelings of guilt, shame, stigmatisation and undesirable or unhelpful past experiences with treatment services contributed to reduced quitting success and acted as an impediment to accessing cessation support. Similarly, a US study of perspectives on smoking cessation treatments among 25 people experiencing homelessness [17] reported a lack of interest in established cessation approaches such as NRT which they viewed negatively. There was a preference to engage in their own self-defined, alternative smoking interventions, including e-cigarettes (EC).

The development and increasing sophistication of EC over the last ten years offers smokers a viable alternative to traditional pharmacotherapies for smoking cessation. There are an estimated 3.6 million current users among the general population in Great Britain [18] and among smokers attempting to quit in England, EC continue to be the most popular method [19]. A In a randomised controlled trial (RCT) of EC versus NRT (participants' choice including combinations) delivered within SSS, 12 month sustained cessation rates were almost doubled in the EC arm: 18 vs 9.9% [20]. Those who opt for EC however, tend to be better educated and higher earners [21]. Although EC are far cheaper than smoking in the longer term, they carry an initial start-up cost which may deter those on lower incomes [22]. In a survey of 283 smokers accessing homeless services across the UK, we found that, although willingness to use

EC was high, only 34% reported that they were willing or able to spend £20 or more on a starter kit [10]. Reducing inequalities in health caused by smoking is a key public health priority [23, 24] and promoting smoking cessation in disadvantaged groups is central to this objective. Although EC have been used in a number of positive trials with smokers in the general population and those with a mental health diagnosis [25] their potential for promoting smoking cessation among the homeless has not been explored.

Nevertheless, the high prevalence of illicit drug and alcohol use and other physical and mental health conditions among those experiencing homelessness [26–28] provides a further challenge for research. Some service providers view smoking as low priority and prioritise treatment for addictions despite evidence that those with a history of other substance use are more likely to die from smoking related disease [29]. Concerns that attempts to quit smoking may also exacerbate other drug use or underlying mental health conditions have also been expressed [30, 31] although recent evidence suggests that this is not the case [32–34]. EC provision in this group could also introduce further challenges around charging, loss and breakage as well as unintended consequences such as increased vulnerability to theft and the use of the device for the administration of other substances [35].

Given the poor outcomes associated with the use of traditional pharmacotherapies among people who experience homelessness who smoke, the reluctance to engage with SSS, and the high initial start-up costs of EC, providing a free EC starter pack at a location already being accessed by this group may help to reduce health inequalities if EC can boost quit rates. Nevertheless, due to the many uncertainties associated with EC use in this population, feasibility work is an important precursor to a definitive trial in order to explore their acceptability and whether smokers experiencing homelessness are willing to engage with the trial procedures.

## Objectives

The overall aim of the research was to undertake a cluster controlled trial to evaluate the feasibility of supplying free EC starter kits for smoking cessation to smokers accessing homeless centres. The following specific objectives were specified.

1. Assess willingness of smokers to participate in the feasibility study to estimate recruitment rates and inform a future trial.

2. Assess participant retention in the intervention and control arms.

3. Assess the potential efficacy of supplying free e-cigarette starter kits to determine the required sample size for a main trial

4. Explore the feasibility of collecting data on contacts with health care services within this population as an input to an economic evaluation in a full RCT

5. Estimate the cost of providing the intervention and usual care.

The study included an embedded qualitative process evaluation in order to examine: perceived facilitators and barriers to engagement; acceptability of the EC intervention and usual care (UC); the impact of local context; and service providers' capacity to support the study. These findings are reported separately.

## Materials and methods

### Study design and setting

This was a four-centre cluster feasibility trial with a nested qualitative process evaluation component (reported separately). Staff administering the intervention (or UC) nor researchers

assessing outcome could be blinded to condition due to the nature of the intervention and study design. Participants were recruited from four homeless centres across the UK: two in London (both residential centres), one in Northampton, and one in Edinburgh (both day centres). Although we had planned to randomise the centres to each condition, actual allocation deviated from protocol due to centre readiness though we balanced potential confounders and differences in environment by ensuring each cluster (EC and UC) contained one day centre and one residential unit.

We allocated the first centre ready to work with us to the EC condition so that we could explore recruitment, 4-week retention and any unintended consequences associated with the intervention to determine whether to proceed with recruitment at centres 2, 3 and 4. Centre 2 (Edinburgh), as the other day centre was therefore allocated to the UC condition. Centres 3 and 4 (London) were allocated to the UC and EC arms, respectively. Centre 3 was allocated to UC as it was geographically closer to the researcher who was still collecting follow up data from centre 1 and we expected lower uptake in the UC condition.

Ethical approval was granted by London South Bank University (LSBU, REF: 1821) and The Salvation Army Ethics Committee. The protocol was published on the NIHR website in September 2018: https://fundingawards.nihr.ac.uk/award/17/44/29

## Participants and recruitment

Eligible participants (see below) were invited for participation by centre staff between 7th January and 5th June 2019. Any smoker interested in using an EC to try to quit smoking or reduce their smoking was eligible to take part. There was no agreement that a cessation attempt should be made and participants did not need to be motivated to quit. Those agreeing to take part were invited to consent and complete a baseline assessment at the homeless centre with a member of the research team at their next visit. Although centres were allocated to condition, participants received the same study information sheet and consent form (where they were informed that they could be allocated to either condition) and completed baseline assessments *before* being told of their condition (though due to the nature of the intervention and centre social dynamics, participants soon became aware of centre allocation). Recruitment and 4 week follow ups ran sequentially across the three sites in England with 12 and 24 week follow ups overlapping. Recruitment and data collection in Edinburgh ran in parallel with the second centre in England.

**Inclusion criteria.**   Aged 18 and over, self-reported daily smoking (confirmed by centre staff), currently accessing homeless centre services and actively engaging with the service (determined by centre staff). In order to represent this population of smokers as accurately as possible, we did not exclude participants on the basis of physical/mental health diagnoses or other use of substances.

**Exclusion criteria.**   Currently using another smoking cessation aid, pregnant, unable to consent (e.g. currently intoxicated or unable to speak English); not known to centre staff.

**Intervention arm–EC starter kit.**   Homeless centre staff saw participants individually and provided them with a starter kit comprising a tank-style refillable EC (worth £20 each) with a choice of: a) nicotine strength e-liquid (2 options: 12 & 18mg/mL) and b) flavours (3 options: tobacco, fruit, menthol). They also received an explanation on how to use the product and a 'guide to e-cigarettes' fact sheet. Participants were given time to try different flavours and nicotine strengths at baseline and be permitted to switch between flavours in accordance with documented vaping practices [36]. Staff also provided participants with five 10ml bottles of e-liquids (£3 each; approx. 7mL a day) for four weeks at weekly intervals in accordance with the upper level reported in the recent UK national survey [37] and encouraged EC charging on site.

**Control arm–Usual care.** Those in the control arm were recruited in the same way as intervention arm (EC) participants and received the same study information sheet and consent form. After meeting the researcher to provide baseline information and being informed they are in the UC arm, participants were referred to their keyworker/other homeless centre staff for an individual appointment to receive brief advice to quit, and a 'help-quit' leaflet (adapted from the NHS Choices website [38]) including information about the location and opening hours of the local SSS. Paper copies of the help-quit leaflet (with SSS contact details) were available as posters/flyers at homeless centres.

To support engagement, all participants were compensated with a £15 voucher for attending each follow-up assessment as this has been shown to improve retention in other studies with homeless smokers [39, 40].

**Staff training and delivery of the intervention.** The research team provided education and training for staff at each centre one to two weeks prior to the beginning of the recruitment period. Training lasted approximately 75 minutes for staff in the UC arm and three hours for those in the EC arm. Content for both groups of staff included information on prevalence and patterns of smoking among the wider population and people experiencing homelessness, the harmful effects of smoking and benefits of quitting. The UC group received information on the role of stop smoking services and how to make a referral to local services. The EC group received information and practical advice on EC, specifically the evidence base of their use among the wider population, effectiveness and safety. Also within the training session, the EC group staff were provided with information about how to deliver correct advice about EC to participants and given a practical hands-on demonstration relating to aspects of EC assembly, use, charging and battery safety. Staff providing EC keyworker sessions were given additional coaching and shadowing opportunities with the researcher.

## Data collection / Measures

**Baseline measures.** • Demographic information and homeless status/history.

- Cigarettes smoked per day (CPD) and in the last seven days, smoking history (e.g. length of smoking, previous number of quit attempts, support used), risky smoking practices (sharing cigarettes, smoking discarded cigarettes, asking strangers for cigarettes) and past and current EC use.

- Severity of tobacco dependence, measured by the Fagerström Test of Cigarette Dependence [41] and expired carbon monoxide (CO).

- Motivation to stop smoking, measured by the Motivation to Stop Scale [42], a 7-level single-item instrument which incorporates intention, desire and belief in quitting smoking.

- Mental health status, measured using the 9-item Patient Health Questionnaire (PHQ-9) [43] for depression (total score ranging from 0 to 27 with a higher score indicating greater severity of depression) and the 7-item Generalised Anxiety Disorder (GAD-7) questionnaire [44] (total score ranging from 0 to 21 with a higher score indicating greater severity of anxiety).

- Alcohol use, measured using the Alcohol Use Disorders Identification Test (AUDIT) [45], a 10-item screening instrument developed by the WHO to screen for a range of drinking problems. Scores range from 0–40 with a score of >8 indicating harmful or hazardous drinking and >13 (females) or >15 (males) indicating alcohol dependence.

- Drug use measured using The Severity of Dependence Scale (SDS) [46], a 5-item screening measure of psychological aspects of dependence. Scores range from 0–15 (low to high

dependence). A 23-item substance use inventory, the RaRE Use of Substances Table [47], recorded types of drugs and substances consumed. For each item, participants indicated frequency of use, 0–6 (Never to Everyday). An added question recorded which substance was the referent for the SDS.

- General health care and service use measured using an adapted health care and social service utilisation questionnaire.

- Health Related Quality of Life (HRQoL) measured using the EQ-5D-3L [48, 49], a generic preference-based measure that consists of a descriptive system and a visual analogue scale (VAS). The descriptive system can be converted to a utility value based on UK population tariff, ranging from condition worth than death (-0.594) to perfect health (1), with 0 represents death [50]. Utility values at multiple time points can be used to derive quality-adjusted life years (QALYs). The EQ VAS represents perceived health on the day of administration, ranging from 0 (death) to 100 (perfect health).

All questionnaires and measures have good psychometric properties and have been used in previous research with vulnerable populations [51–53].

Following completion of these baseline measures, participants were informed of their allocated condition and then met with their keyworker or other centre staff who provided the EC starter kit or UC (according to homeless centre allocation). A follow up appointment was then made, and in order to minimise attrition (where possible and where consent was given) mobile telephone numbers and email addresses were recorded in order to send text message reminders ahead of scheduled appointments.

**Follow up data collection.** Follow up data was collected at weeks 4, 12 and 24 between 4th February and 20th November 2019. The following information was collected: self-reported smoking abstinence; number of cigarettes smoked; breath CO levels; engagement with the local SSS; use of EC and other tobacco/nicotine containing products; general health care and service use; HRQoL (EQ-5D-3L); mental health status (GAD-7, PHQ-9); other drug use/ dependence (AUDIT, SDS); direct and indirect staff contact time. For those in the EC arm, we additionally measured 12 positive (e.g. throat hit, satisfaction, pleasant, craving reduction) and 21 negative effects (e.g. mouth/throat irritation, nausea, headache, heartburn) of EC use using a Visual Analogue Scales (VAS) and summed to create a percentage score (higher score = higher positive or negative effect) as used in our previous studies [54, 55]. To further monitor risk and adverse effects (e.g. EC theft, exchanges, use for other substances), we utilised a purposefully developed unintended consequences checklist. A £15 love to shop voucher was provided for each follow up.

## Feasibility outcome measures

1. To assess willingness to take part in the trial, we recorded the number of people who were asked, and the number who consented, to take part.

2. Retention and engagement was measured by recording a) the proportion of participants: i) still using EC in the intervention arm and ii) who had visited the SSS in the UC arm; and b) the proportion of participants who completed assessment measures in each arm at each time point.

3. To assess the potential efficacy of supplying free EC starter kits, at each follow up point, we recorded: a) CO-validated (<8ppm) sustained abstinence (from 2-weeks post-quit date allowing up to 5 slips; the gold 'Russell Standard' in smoking cessation research) [56]; b)

CO-validated 7-day point prevalence abstinence (i.e. no smoking at all in the last 7 days), the most commonly reported outcome measure in studies of smoking cessation in people experiencing homelessness [6]; and c) the proportion achieving 50% smoking reduction, a common outcome measure used in EC cessation studies [57] (calculated by subtracting CPD at follow up from baseline).

4. To explore the feasibility of collecting data on contacts with health care services we recorded participant utilisation of primary and secondary health care services using a self-report questionnaire

5. Staff contact time, non-contact time and other resources used in the delivery including staff costs, e-cigarettes and other costs incurred was collected to provide an indicative cost of the intervention.

## Sample size

As this is a feasibility study, a formal power calculation based on detecting evidence for efficacy was not conducted; indeed, an aim of the feasibility study was to calculate the required sample size (and an intra class correlation coefficient (ICC) for a possible future definitive cluster RCT. Our original recruitment goal was 120 within a 5 month period; that is, approximately 30 per centre, with sampling from centres taking place consecutively. This was based on our preliminary scoping work suggesting that each homeless centre has contact with between 25 and 120 homeless clients every day of whom 70–90% are likely to be smokers. Other studies in homeless populations have reported follow-up rates ranging between 24% and 88% (depending on the location of visits, provision of incentives & use of prompts, see Richards et al. 2015 [40]). Therefore, estimating that 50% of those who agree would drop out in the period between consenting to participate and the final follow up at 24 weeks, the sample size at the final follow up was estimated to be a minimum of 60. This was a pragmatically chosen sample size, based on the information available to us at the time, to allow us to identify evidence of feasibility, recruitment rates and any problems with the intervention or research methods.

## Data analysis

Our feasibility data were analysed according to our protocol https://fundingawards.nihr.ac.uk/award/17/44/29. To address objectives 1 and 2 willingness to participate and retention, we present frequency information regarding the number of eligible smokers who: i) were invited to take part; ii) consented/completed the baseline assessment; iii) attended and completed each follow up; and iv) were still engaging with the treatment/UC at each follow up.

Baseline demographic information is summarised using frequencies and descriptive statistics and the arms (EC v UC) are compared using t-tests, Mann- Whitney U tests or Fisher's Exact test as appropriate. EC effects are summarised descriptively.

To assess the potential efficacy of supplying free EC starter kits and to inform the sample size for a future larger trial (objective 3), we summarise the proportion of participants reporting sustained smoking abstinence (CO verified), 7-day point prevalence abstinence, and a 50% reduction in smoking in each arm at each follow up time point. The main sample size calculation is based on 24 week sustained abstinence using intention to treat analysis; that is, all those randomised are included in the analysis as belonging to the group to which they were randomised and those with missing outcome data were treated as smokers. The planned analysis was to use generalised linear mixed effects models to estimate the clustering of observations within centres. From these models we would also estimate the effect size of the intervention after adjustment for covariates. The ICC was to be estimated from the random intercept model.

To explore feasibility of collecting data using self-report questionnaire, we administered EQ-5D-3L and healthcare and social service questionnaire to participants and examined the completeness of the responses and the frequency of each service use. We presented QALYs derived from EQ-5D-3L using the area under the curve approach [58] and costs of general healthcare services at each time point, by arm, to show a preliminary profile of costs and HRQoL. To provide an indicative cost of intervention, we estimated the costs of pre-intervention training and intervention delivery, based on relevant staff activities.

For the primary outcome, participants lost to follow up were assumed to be smoking and included at follow up. For assessment of the relationship between other variables and outcomes, cases with missing data were excluded from the analysis. Data were analysed in SPSS and STATA.

## Results

### Recruitment

One hundred and seventy seven participants were invited to participate (106 in the EC arm and 71 in the UC arm). Of these, 24 were not eligible (16 in the EC arm; 8 in the UC arm). Reasons for exclusion were: persons presenting were non-smokers, pregnant, unknown to centre staff, assessed to be unable to provide informed consent due to levels of intoxication, or assessed by staff to have high mental health risk or self-care burden. Of the 153 eligible participants, 80 consented, completed baseline assessments and were randomised: 48 (53%) in the EC arm and 32 (51%) in the UC arm. Fig 1 shows the CONSORT flow diagram and recruitment and retention figures by centre can be found in S1 Table.

### Demographic and smoking related characteristics

As presented in Table 1, the mean age for the total sample was 42.66 years and 65% were male. Participants were primarily white (76.3%), heterosexual (85%) and 37.6% were educated to A-level (or equivalent) or higher. Employment status varied with only 2.5% currently in paid employment and 97.5% were in receipt of public funds (welfare benefits). Just over half of the sample (60%) were currently housed in supported accommodation or in a hostel. Seventy-four percent of the sample reported a long-standing illness, 38.8% had been admitted to hospital due to mental illness and 50% had previously spent time in prison. The mean number of cigarettes smoked per day was 20 and most participants reported that they had started smoking during their teenage years (M = 15 years). Baseline expired CO was 20.29 ppm and the mean FTCD score was 5.51. Fifty five percent reported that they shared cigarettes, 43% reported that they had smoked discarded cigarettes and 40% had asked strangers for cigarettes at least occasionally. Motivation to stop smoking (MTSS) varied considerably although only 6.3% reported that they did not want to stop smoking. Mean scores on the GAD-7 (11.22) and PHQ-9 (12.93) were in the moderate/moderately severe range for anxiety and depression and the mean AUDIT score (9.22) suggests that on average, participants were drinking at harmful/hazardous levels. The EC and UC arms differed on several variables: the proportion of participants who had previously spent time in prison or who had a long-standing illness, disability or infirmity was significantly higher in the UC than EC arm. UC arm participants also scored higher on anxiety and substance dependence and were less motivated to quit smoking (see Table 1).

### Follow up and retention

Retention rates (percentage of those allocated) were 75%, 65% and 59% respectively at 4, 12 and 24 weeks and retention was higher in the EC arm. Loss to follow up occurred mainly

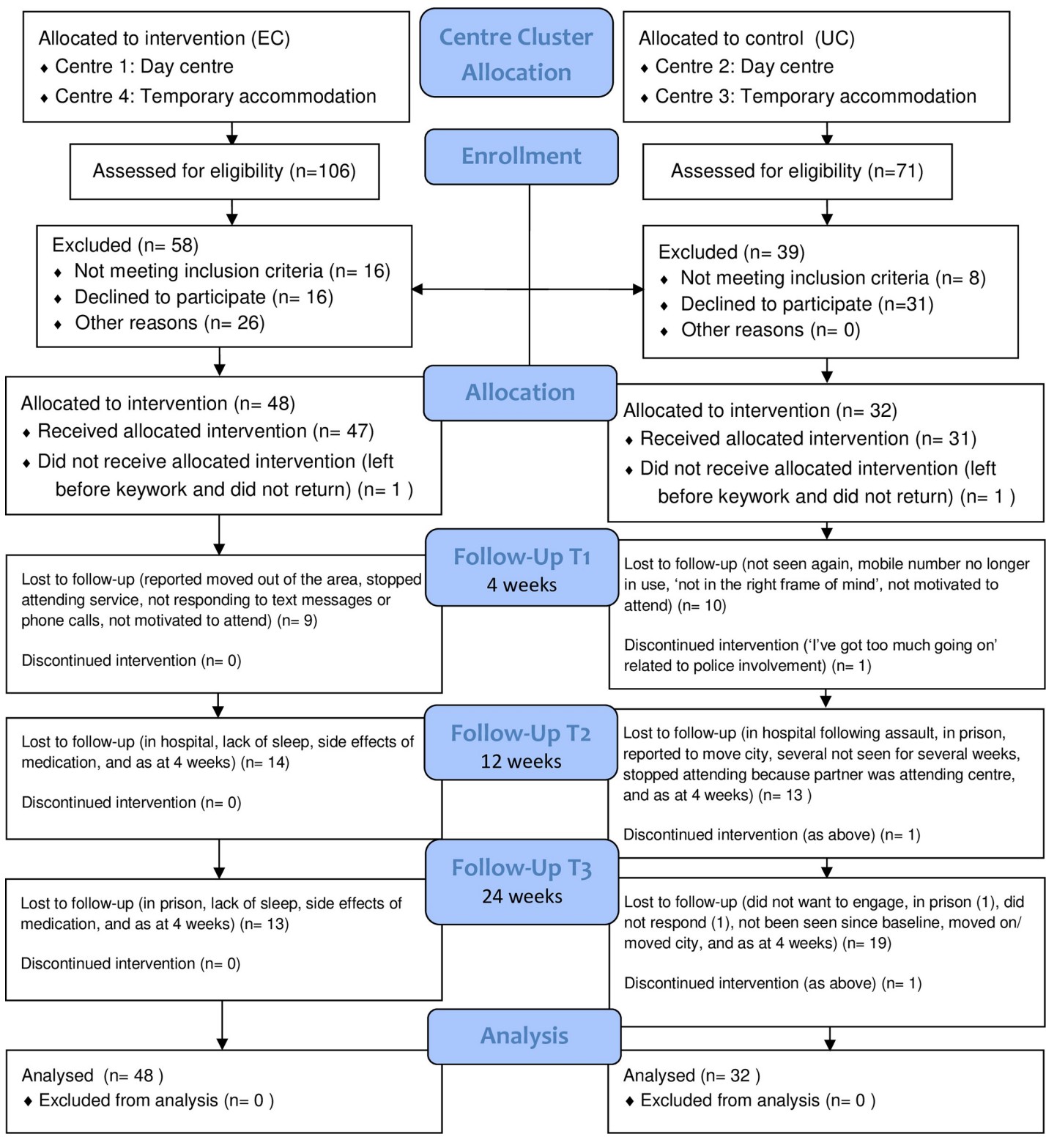

**Fig 1. CONSORT flow diagram.**

**Table 1. Participant baseline characteristics.**

| | Total N = 80 | EC arm | UC arm | P value |
|---|---|---|---|---|
| Age (in years): mean (SD) | 42.66 (10.79) | 42.75 (10.90) | 42.53 (10.78) | .93 |
| Gender: N (%) | | | | .34 |
| Female | 28 (35) | 19 (40) | 9 (28.13) | |
| Male | 52 (65) | 29 (60) | 23 (71.88) | |
| Employment status N (%) | | | | .11 |
| Full time school or college | 0 (0) | 0 (0) | 0 (0) | |
| Paid employment/self-employed | 2 (2.5) | 2 (4.16) | 0 (0) | |
| Government training scheme | 2 (2.5) | 1 (2.08) | 1 (3.13) | |
| Unpaid or voluntary work | 9 (11.3) | 9 (18.75) | 0 (0) | |
| Waiting to work, already obtained | 1 (1.3) | 1 (2.08) | 0 (0) | |
| Looking for work/training scheme | 12 (15.0) | 7 (14.58) | 5 (15.63) | |
| Prevented by temporary sickness/injury | 5 (6.3) | 2 (4.16) | 3 (9.38) | |
| Permanently unable to work | 38 (47.5) | 21 (43.75) | 17 (53.13) | |
| Unemployed and not looking for work | 5 (6.3) | 3 (6.25) | 2 (6.25) | |
| Other | 6 (7.5) | 2 (4.16) | 4 (12.5) | |
| Current sleeping situation (last 7 days): N (%)* | | | | |
| Sleeping rough on streets/park | 7 (8.8) | 3 (6.25) | 4 (12.5) | .42 |
| Hostel or supported accommodation | 48 (60) | 31 (64.58) | 17 (53.13) | .36 |
| Sleeping on somebody's floor/sofa | 3 (3.8) | 1 (2.08) | 2 (6.25) | .56 |
| Emergency accommodation (refuge, shelter) | 9 (11.3) | 8 (16.66) | 1 (3.13) | .08 |
| B&B or temporary accommodation | 2 (2.5) | 1 (2.08) | 1 (3.13) | 1.00 |
| Housed–own tenancy | 18 (22.5) | 8 (16.66) | 10 (31.25) | .17 |
| Other | 2 (2.5) | 2 (4.16) | 0 (0) | .51 |
| Backgrounds: N (%)* | | | | |
| Spent time in prison | 40 (50) | 19 (40) | 21 (65.63) | **.04** |
| Spent time in secure/young offender unit | 18 (22.5) | 9 (18.75) | 9 (28.13) | .41 |
| Spent time in local authority care | 17 (21.3) | 7 (14.58) | 10 (31.25) | .10 |
| Spent time in the armed forces | 7 (8.8) | 3 (6.25) | 4 (12.5) | .43 |
| Admitted to hospital due to mental illness | 31 (38.8) | 18 (37.5) | 13 (40.63) | 1.00 |
| Been a victim of domestic violence | 31 (38.8) | 18 (37.5) | 13 (40.63) | 1.00 |
| Highest level of education: N (%) | | | | .40 |
| School (stopped prior to GCSE/standard grade | 24 (30) | 17 (35.42) | 7 (21.88) | |
| School (GCSE/Standard grade) | 26 (32.5) | 13 (27.08) | 13 (40.63) | |
| College (A-level/FE/Highers) | 25 (31.3) | 15 (31.25) | 10 (31.25) | |
| University (degree level) | 4 (5) | 3 (6.25) | 1 (3.13) | |
| University (post-graduate, higher level) | 1 (1.3) | 0 (0) | 1 (3.13) | |
| Ethnicity: N (%) | | | | .64 |
| White | 61 (76.3) | 35 (72.92) | 26 (81.25) | |
| Asian/Asian British | 2 (2.6) | 6 (12.50) | 2 (6.25) | |
| Black/Black British | 9 (11.4) | 2 (4.17)) | 0 (0) | |
| Mixed race/multiple ethnic groups | 8 (10.2) | 5 (10.42) | 4 (12.5) | |
| Sexual Orientation: N (%) | | | | .81 |
| Heterosexual or straight | 68 (85) | 40 (83.33) | 28 (87.5) | |
| Gay or lesbian | 1 (1.3) | 1 (2.08) | 0 (0) | |
| Bi-sexual | 3 (3.8) | 2 (4.16) | 1 (3.13) | |
| Prefer to self-define | 3 (3.8) | 2 (4.16) | 1 (3.13) | |
| Prefer not to say | 3 (3.8) | 3 (6.25) | 0 (0) | |
| Missing | 2 (2.5) | 0 (0) | 2 (6.25) | |

(*Continued*)

**Table 1.** (Continued)

| | Total N = 80 | EC arm | UC arm | P value |
|---|---|---|---|---|
| Immigration Status: N (%) | | | | .68 |
| UK National | 74 (92.5) | 45 (93.75) | 29 (90.63) | |
| European Economic Area (EEA) national | 6 (7.5) | 3 (6.25) | 3 (9.38) | |
| Receiving public Funds (Benefits): N (%) | | | | .16 |
| Yes | 78 (97.5) | 48 (100) | 30 (93.75) | |
| No | 2 (2.5) | 0 (0) | 2 (6.25) | |
| Long-standing illness, disability, infirmity: N (%) | | | | **.02** |
| Yes | 59 (73.8) | 30 (62.5) | 29 (90.63) | |
| No | 16 (20) | 13 (27.08) | 3 (9.38) | |
| Prefer to self-define | 4 (5) | 4 (8.33) | 0 (0) | |
| Prefer not to say | 1 (1.3) | 1 (2.08) | 0 (0) | |
| Number of cigarettes/day: mean (SD) | 20.07 (15.33) | 20.5 (16.78) | 19.41 (13.07) | .86 |
| Expired CO: mean (SD) | 20.29 (10.04) | 19.60 (9.58) | 21.31 (10.77) | .46 |
| FTCD: mean (SD) | 5.51 (2.47) | 5.24 (2.53) | 6.13 (2.35) | .12 |
| Age started smoking: mean (SD) | 15.17 (5.47) | 16.02 (6.30) | 13.92 (3.72) | .09 |
| Sharing cigarettes: N (%) | | | | .10 |
| Not at all | 35 (43.8) | 25 (52.08) | 9 (28.13) | |
| Occasionally | 18 (22.5) | 10 (20.83) | 7 (21.88) | |
| Regularly | 7 (8.8) | 3 (6.25) | 5 (15.63) | |
| Daily | 19 (23.8) | 9 (18.75) | 11 (34.38) | |
| Smoke discarded cigarettes: N (%) | | | | .43 |
| Not at all | 45 (56.3) | 30 (62.5) | 15 (46.88) | |
| Occasionally | 22 (27.5) | 12 (25) | 10 (31.25) | |
| Regularly | 7 (8.8) | 3 (6.25) | 4 (12.5) | |
| Daily | 5 (6.3) | 2 (4.16) | 3 (9.38) | |
| Ask strangers for cigarettes: N (%) | | | | .35 |
| Not at all | 47 (58.8) | 31 (64.58) | 16 (50) | |
| Occasionally | 21 (26.3) | 12 (25) | 9 (28.13) | |
| Regularly | 5 (6.3) | 2 (4.16) | 3 (9.38) | |
| Daily | 6 (7.5) | 2 (4.16) | 4 (12.5) | |
| MTSS: N (%) | | | | **.04** |
| I don't want to stop smoking | 5 (6.3) | 1 (2.08) | 4 (12.5) | |
| I think I should stop but don't really want to | 8 (10) | 4 (8.33) | 4 (12.5) | |
| I want to stop but haven't thought about when | 14 (17.5) | 6 (12.5) | 4 (12.5) | |
| I really want to stop but I don't know when I will | 11 (13.8) | 3 (6.25) | 8 (25) | |
| I want to stop smoking and hope to soon | 18 (22.5) | 16 (33.33) | 5 (15.63) | |
| I really want to stop & intend to within 3 months | 7 (8.8) | 5 (10.42) | 3 (9.38) | |
| I really want to stop and intend to within 1 month | 14 (17.5) | 11 (22.92) | 3 (9.38) | |
| Missing | 3 (3.8) | 2 (4.16) | 1 (3.13) | |
| Importance of quitting at this attempt: N (%) | | | | .25 |
| Desperately important | 13 (16.3) | 10 (20.83) | 3 (9.38) | |
| Very important | 38 (47.5) | 25 (52.08) | 15 (46.88) | |
| Quite important | 19 (23.8) | 7 (14.58) | 10 (31.25) | |
| Not at all important | 8 (10) | 5 (10.42) | 3 (9.38) | |

(*Continued*)

**Table 1.** (Continued)

|  | Total N = 80 | EC arm | UC arm | P value |
|---|---|---|---|---|
| Determination to quit at this attempt: N (%) |  |  |  | .52 |
| Extremely determined | 19 (23.8) | 13 (27.08) | 6 (18.75) |  |
| Very determined | 26 (32.5) | 16 (33.33) | 10 (31.25) |  |
| Quite determined | 24 (30.5) | 14 (29.16) | 10 (31.25) |  |
| Not at all determined | 8 (10.0) | 3 (6.25) | 5 (15.63) |  |
| Missing | 3 (3.8) |  |  |  |
| Self-rated chance of quitting (1 = very low– 6 = extremely high): median (IQR) | 4 (2) | 4(1) | 4 (1) | .23 |
| GAD-7: median (IQR) | 11 (13 | 10 (11) | 16 (13) | **.03** |
| PHQ-9: median (IQR) | 14 (14.5) | 13.5 (13) | 15.5 (16) | .45 |
| AUDIT: median (IQR) | 5.5 (13) | 5 (17) | 6.5 (11) | 1.0 |
| SDS: median (IQR) | 3 (5.5) | 2 (7) | 8 (8) | **< .01** |

*Participants could select more than one option. MTSS = Motivation to Stop Smoking Scale: GAD-7 = Generalised Anxiety Disorder Questionnaire: PHQ-9 = Patient Health Questionnaire; AUDIT = Alcohol Use Disorders Identification Test; SDS = Severity of Dependence Scale

T-tests/Mann Whitney U used for continuous variables. Fishers Exact Test used for categorical variables.

between 12 and 24 weeks in the UC arm primarily because participants were no longer attending the homeless service. The flow of participants through the study is shown in the CONSORT diagram in Fig 1 and retention rates at each centre are shown in S1 Table.

Of those participants in the EC arm who attended follow ups and answered the question: 33/39 (85%), 28/34 (82%) and 22/35 (63%) reported that they still had the e-cigarette that we provided at 4, 12 and 24 weeks respectively. The number of participants who reported that they were still using an e-cigarette (either the one we provided or a different one) was 37/39 (95%), 30/33 (91%) and 27/34 (79%) at 4, 12 and 24 week respectively. In the UC arm, 4/16 (25%) participants who attended the 4 week follow up reported that they had followed the recommendation to attend the SSS. No additional participants attended between weeks 4 and 24 (although 2 of the 4 initial attendees reporting attending more than once).

## E-cigarette use, effects and unintended consequences in the EC arm

At 4 weeks, of those attending follow up appointments (N = 39) three (8%) participants reported that they had lost their e-cigarette, three (8%) reported that it had been stolen, one (3%) had swapped it and 14 (36%) reported that it had broken (see Table 2). The most frequently reported reason for breakage was that the e-cigarette had been dropped and the glass tank broke. Three of 39 (8%) participants reported that they had purchased their own EC at 4-weeks, 4/34 (12%) at 12 weeks and 11/35 (31%) at 24 weeks. No participants reported adding any other substance to their e-cigarette at any follow up time point although two reported that they had previously added THC or CBD oil to a different device. Self-reported negative effects were rare with a mean percentage score of 13.73 (12.95) at 4 weeks (Table 3; data from 12 and 24 weeks are not presented as this fell beyond the period of our e-liquid provision but are available upon request). The highest scoring negative effect was 'nervous' (M = 19.97 SD = 29.01) followed by 'headache' (M = 18.54, SD = 28.49). In terms of positive effects (Table 3), the mean total percentage score was 49.51 (18.39). The highest scoring positive effects were 'pleasant' (M = 73.79, SD = 22.60) and 'tastes good' (71.92, SD = 25.21) and the lowest scoring positive effect was 'tastes like my usual brand' (M = 19.97, SD 29.85).

**Table 2. E-cigarette events at 4, 12 and 24 weeks.**

| 4-weeks: | YES | |
|---|---|---|
| | **N** | **%** |
| E-cig lost: N (%) | 3/39 | 8 |
| E-cig stolen: N (%) | 3/39 | 8 |
| Sold: N (%) | 0/39 | 0 |
| Exchanged/swapped: N (%) | 1/39 | 3 |
| Given away N (%) | 0/39 | 0 |
| Broken: N (%) | 14/39 | 36 |
| Added substance?: N (%) | 0/39 | 0 |
| **12 weeks**[*] | | |
| E-cig lost (%) | 2 /32 | 6 |
| E-cig stolen | 1/32 | 3 |
| Sold | 0/32 | 0 |
| Exchanged/swapped | 0/32 | 0 |
| Given away | 1/32 | 3 |
| Broken | 11/31 | 35 |
| Added substance? | 0/34 | 0 |
| **24 weeks**[*] | | |
| E-cig lost (%) | 2/33 | 6 |
| E-cig stolen | 2/33 | 6 |
| Sold | 0/33 | 0 |
| Exchanged/swapped | 0/33 | 0 |
| Given away | 2/33 | 6 |
| Broken | 11/33 | 33 |
| Added substance? | 0/34 | 0 |

N = number reporting / number who attended that session

[*] Although participants were asked "since the last meeting", some of the 12 and 24 week responses relate to breakages, loss etc. from the previous time point so these cannot be considered additional breakages, losses etc. Some participants reported exchanging or giving away devices because they broke. Some participants reported replacing devices which subsequently broke or were lost, stolen etc.

## Mental health and substance use

Mean (SD) scores for the mental health and substance use measures for those participants who were retained in the study through to the 24 week follow up are presented in Table 4. Both GAD-7 anxiety and PHQ-9 depression scores showed a steadily decline from baseline to the 24 week follow up. A similar decline in AUDIT scores was also evident although SDS remained stable.

## Smoking outcome data

Table 5 presents the smoking related outcome measures. Of those who could be followed up, the CO validated sustained abstinence rate at 24 weeks was 3/35 (11%) for the EC arm and 0/12 (0%) for the UC arm. Assuming that all those with missing follow up data were smoking (intention to treat), the 24 week sustained abstinence rate was 6.25% (3/48) in the EC arm vs. 0/32 (0%) in the UC arm. Seven day point prevalence rates at 24 weeks were the same as sustained abstinence rates. The percentage of participants who reported >50% reduction in CPD from baseline to 24 week follow up was 43% in the EC arm and 25% in the UC arm. The

**Table 3. Self-reported negative and positive effects at week 4 in the E-cigarette arm.**

| Negative Effect | Mean (SD) | Positive Effect | Mean (SD) |
|---|---|---|---|
| Nervous | 19.97 (29.01) | Pleasant | 73.79 (22.60) |
| Headache | 18.54 (28.49) | Tastes good | 71.92 (25.21) |
| Sweaty | 18.46 (22.73) | Satisfying | 70.56 (26.84) |
| Weak | 18.13 (21.56) | Craving | 69.49 (30.66) |
| Nausea | 17.38 (25.45) | Calmer | 53.38 (33.50) |
| Heart pounding | 17.36 (28.30) | Nicotine hit | 52.05 (36.95) |
| Light headed | 14.67 (20.55) | Throat hit | 47.77 (34.48) |
| Confused | 14.21 (21.94) | Concentration | 46.08 (37.37) |
| Dizzy | 14.10 (21.66) | Awake | 37.95 (36.84) |
| Hiccups | 14.00 (23.53) | Hunger | 27.95 (31.83) |
| Stomach ache | 13.59 (20.89) | Feels like usual brand/model | 23.15 (32.18) |
| Salivation | 13.46 (21.40) | Tastes like usual brand/model | 19.97 (29.85) |
| Mouth irritation | 13.38 (22.25) | | |
| Flatulence/bloating | 13.36 (23.65) | | |
| Throat irritation | 12.49 (17.54) | | |
| Cold hands/feet | 11.31 (22.04) | | |
| Palpitations | 10.44 (15.91) | | |
| Aching jaw | 9.49 (20.42) | | |
| Heartburn | 9.26 (12.88) | | |
| Diarrhoea | 9.13 (16.46) | | |
| Vomiting | 5.59 (9.66) | | |
| TOTAL | 13.73 (12.95) | TOTAL | 49.51 (18.39) |

(N = 39) Scores are expressed as a percentage converted from Visual Analogue Scales.

percentage of those with >50% reduction in expired CO was a little lower and similar in the two groups (EC = 20%; UC = 25%). The number of people reporting that they shared cigarettes remained relatively stable across the 24 weeks although reports of smoking discarded cigarettes and asking strangers for cigarettes decreased (data in S2 Table).

## Intraclass correlation coefficient estimation

The planned analysis included the fitting of a binary logistic mixed effects models for the purpose of estimating the intraclass correlation coefficient based on the random intercept. It was not possible to achieve convergence for the mixed effects model within the attained sample so the intraclass correlation coefficient is calculated by the Fleiss-Cuzick method [59] and the point estimate is 0.0157. The intra class correlation coefficient is the 'proportion of the total variance in the outcome attributable to variance between clusters' [59]. Although the estimated ICC is relatively small [60] neglecting to factor this in to the sample size calculation for a full scale cluster randomised controlled trial would result in reduced statistical power.

## E-cigarette use in the UC arm

Four of the 21 UC participants (19%) tested at the 4-week follow up reported that they had purchased an EC. At 12 and 24 weeks respectively a further 1/18 (5.6%) and 2/12 (16.7%) reported that they had made an EC purchase. Any use of an EC during the study period was reported by 9/14 (64%) participants at 4 weeks, 5/18 (28%) at 12 weeks and 4/12 (33%) participants at 24 weeks.

**Table 4. Mental health and substance use at baseline and each follow up time point for the e-cigarette (EC) and usual care (UC) arms.**

|  | Total | EC Arm | UC Arm |
|---|---|---|---|
| **GAD-7** |  |  |  |
| N | 37 | 27 | 10 |
| Baseline: Mean (SD) | 10.08 (6.91) | 8.48 (6.81) | 14.40 (5.36) |
| 4 weeks: Mean (SD) | 9.11 (6.81) | 7.15 6.38) | 14.40 (5.02) |
| 12 weeks: Mean (SD) | 8.27 (7.12) | 6.44 (6.54) | 13.20 (6.49) |
| 24 weeks: Mean (SD) | 7.54 (6.64) | 5.63 (6.34) | 12.70 (4.42) |
| **PHQ-9** |  |  |  |
| N | 36 | 25 | 11 |
| Baseline: Mean (SD) | 11.53 (8.26) | 10.76 (8.34) | 13.27 (8.17) |
| 4 weeks: Mean (SD) | 10.53 (7.62) | 9.08 (7.76) | 13.82 (6.46) |
| 12 weeks: Mean (SD) | 9.92 (8.03) | 8.36 (7.53) | 13.45 (8.37) |
| 24 weeks: Mean (SD) | 8.25 (7.38) | 7.12 (7.22) | 10.82 (7.23) |
| **AUDIT** |  |  |  |
| N | 35 | 24 | 11 |
| Baseline: Mean (SD) | 10.60 (11.07) | 11.83 (12.15) | 7.91 (8.08) |
| 4 weeks: Mean (SD) | 7.86 (9.34) | 9.08 (9.76) | 5.18 (8.48) |
| 12 weeks: Mean (SD) | 8.57 (9.72) | 10.29 (10.37) | 4.82 (7.15) |
| 24 weeks: Mean (SD) | 8.00 (8.47) | 8.92 (8.83) | 6.00 (7.64) |
| **SDS** |  |  |  |
| N | 37 | 27 | 10 |
| Baseline: Mean (SD) | 3.81 (5.05) | 3.15 (4.33) | 5.50 (6.52) |
| 4 weeks: Mean (SD) | 3.19 (4.48) | 2.81 (4.51) | 4.20 (4.49) |
| 12 weeks: Mean (SD) | 3.78 (4.15) | 3.35 (4.05) | 4.90 (4.41) |
| 24 weeks: Mean (SD) | 3.78 (4.33) | 3.23 (4.46) | 5.20 (3.82) |

GAD-7 = Generalised Anxiety Disorder Questionnaire: PHQ-9 = Patient Health Questionnaire; AUDIT = Alcohol Use Disorders Identification Test; SDS = Severity of Dependence Scale

## Data collection for an economic evaluation

Missing information on both EQ-5D-3L and general healthcare services questionnaire was largely due to participants' absence at the time of interviews. For those who attended the interview, there were few cases of items missing.

Amongst observed cases, the proportion of participants scoring no problem on EQ-5D-3L was consistently higher in the EC arm than the UC arm across all domains and all time points. Consequently, mean utility scores in the EC arm were higher amongst those competing the measure (Table 6). For those completing EQ-5D-3L at all time points, mean QALY was 0.195 (SD 0.097) in the UC arm (n = 11) and 0.315 (SD 0.120) in the EC arm (n = 26), over 24 weeks.

Compared with the EC arm, the UC arm showed an indication of relying more on SSS and NRT. When the EC arm were no longer offered sessions from keyworkers, they turned to GPs, practice nurses or pharmacists for help (see S3 Table). No use of NHS SSS helplines was reported throughout the study period.

Community-based services (maternity service, sex health clinic, early intervention team, detox/rehab unit) were rarely used in either arm. Other services such as drug and alcohol service, adult mental health team and housing team were used more often. While GP and practice

**Table 5. Frequencies and percentages for smoking related outcome variables at baseline and each follow up time point for the e-cigarette (EC) and usual care (UC) arms.**

|  | EC Arm | UC Arm |
|---|---|---|
| **Sustained abstinence** |  |  |
| 4 weeks: N (%) | 8/39 (21%) | 0/21 (0%) |
| 12 weeks: N (%) | 2/34 (6%) | 0/18 (0%) |
| 24 weeks: N (%) | 3/35 (11%) | 0/12 (0%) |
| **7 day point prevalence** |  |  |
| 4 weeks | 7/39 (18%) | 0/21 (0%) |
| 12 weeks | 5/34 (15%) | 0/18 (0%) |
| 24 weeks | 3/35 (9%) | 0/12 (0%) |
| **50% reduction in CPD** |  |  |
| 4 weeks | 21/39 (54%) | 4/21 (19%) |
| 12 weeks | 20/34 (59%) | 4/18 (22%) |
| 24 weeks | 15/35 (43%) | 3/12 (25%) |
| **50% reduction in expired CO** |  |  |
| 4 weeks | 10/39 (26%) | 2/21 (10%) |
| 12 weeks | 7/34 (21%) | 1/18 (6%) |
| 24 weeks | 7/35 (20%) | 3/12 (25%) |

N = number meeting criteria / number attending the session

nurse services were used more frequently, home visits were rare. The majority of participants in either arm received at least one prescription at each time point. Accident and Emergency (A & E) and hospital visits were reported in both arms at all time points.

The mean costs among observed cases were consistently higher in the UC arm than in the EC arm (Table 6). The difference up to 12 weeks was mostly due to the longer hospital stay in the UC arm. At 24 weeks, although the mean cost of hospital stay and drug/alcohol service in the EC arm was much higher, the mean cost of A & E, early intervention team and adult mental health team offset the reducing costs in other services in the UC arm.

**Table 6. Mean EQ-5D-3L utility value, EQ VAS and costs of general healthcare and social services at each time point, by arm.**

|  | UC (n = 32) | | | EC (n = 48) | | |
|---|---|---|---|---|---|---|
|  | n | Mean (SD) | | n | Mean (SD) | |
| **EQ-5D-3L** |  |  |  |  |  |  |
|  |  | Utility | EQ VAS |  | Utility | EQ VAS |
| Baseline | 31 | 0.394 (0.362) | 57.0 (21.7) | 46 | 0.548 (0.341) | 52.7 (20.7) |
| 4 weeks | 21 | 0.330 (0.308) | 48.6 (24.7) | 39 | 0.602 (0.346) | 61.4 (21.5) |
| 12 weeks | 18 | 0.350 (0.351) | 59.3 (22.7) | 33 | 0.683 (0.309) | 65.6 (21.3) |
| 24 weeks | 12 | 0.619 (0.238) | 61.0 (22.5) | 34 | 0.653 (0.363) | 61.8 (21.6) |
| **Costs of general healthcare and social services** |  |  |  |  |  |  |
|  |  | Costs |  |  | Costs |  |
| Baseline | 31 | £1,480 (£3,188) |  | 47 | £518 (£754) |  |
| 4 weeks | 21 | £1,559 (£4,489) |  | 39 | £539 (£987) |  |
| 12 weeks | 18 | £957 (£1,849) |  | 34 | £682 (£885) |  |
| 24 weeks | 12 | £1,207 (£1,494) |  | 35 | £1,172 (£1,952) |  |

## Costs of the intervention

The costs of staff time in delivering EC training was £1,041 in total and UC training was £247. The costs of staff time in attending the training were £803 in the EC arm and £185 in the UC arm. The costs of e-cigarette and e-liquid used were £138. The pamphlets for EC training were printed at £8 in total and for UC training at £2.40 in total. In EC training, six e-cigarette devices were given to the centre staff who smoke, costing £120. The costs of inter-city travelling, travelling time, accommodation and meals were estimated at £294 in the EC arm and £1,143 in the UC arm. In total, the training costs for the UC arm were £1,577 and for the EC arm £2,403. Allocating evenly to participants in each arm, it was £49.27 per participant in the UC arm and £50.07 per participant in the EC arm. Due to the staff capability, the data on delivery sessions were not collected in the UC arm. We therefore did not know the attendance of weekly sessions (visit 0–3).

For the EC arm, 47 out of 48 participants attended visit 0 with staff and received an EC starter kit and five bottles of e-liquid (235 bottles in total), 46 of which also received pamphlets of e-cigarette device instruction with one missing. At visits 1, 2 and 3, an additional 147, 114 and 110 bottles of e-liquid were supplied totalling 606 across the four weeks (see S4 Table for a breakdown of flavours and nicotine strengths supplied at each time point). Except for missing or absent at visit, the mean duration of visit 0 was 25 minutes (SD 16 minutes, range 7–80, n = 46), of visit 1 was 8 minutes (SD 4 minutes, range 1–20, n = 29), of visit 2 was 6 minutes (SD 3 minutes, range 1–15, n = 21), and of visit 3 was 5 minutes (SD 2 minutes, range 2–10, n = 20). Visits 1–3 were fairly short as they mainly involved supplying further bottles of e-liquid.

The mean costs of EC starter kit and e-liquid were £57.46 (SD £19.91) among the 48 participants in the EC arm. The mean cost of pamphlet was £0.39 (SD £0.06) among 47 participants in the arm. The mean costs of sessions were £7.51 (SD £4.60) among 43 participants. The costs of intervention delivery were therefore estimated for 43 participants whose data were complete in this part and it was £64.35 (SD £22.89) per participant.

In total, the mean costs of EC intervention, including training and delivery, were estimated to be £114.42 (SD £22.89) among 43 participants in the EC arm, with five participants with incomplete information.

## Main RCT sample size calculation

Based on the proportions with sustained CO validated abstinence in each arm (6.25% in intervention 0% in usual care) and assuming 0.05 alpha, 90% power and intraclass correlation coefficient of 0.01, 12 participants per cluster (the feasibility study average) a full trial would require 16 sites per arm, and the study would need to enrol 192 participants per arm, 384 participants in total. ICC is estimated at 0.01 as it is anticipated the inclusion of individual level predictors will further reduce the ICC from the estimate calculated without adjustment.

This sample size estimate assumes equal cluster sizes. These estimates are not adjusted for attrition as it is anticipated that the primary analysis will be by intention to treat and participants lost to follow up assumed to have relapsed entailing no loss of power.

## Discussion

In the first attempt worldwide to explore supplying free EC starter kits to smokers accessing homeless centres, this feasibility trial captured data on participant recruitment, retention, engagement and preliminary efficacy and cost-effectiveness.

In terms of willingness to participate, just over half of eligible participants invited, consented to take part and completed baseline assessments. The study also provided useful

information and insights into the types of homeless centres to include in a future main cRCT; the number of eligible participants varied considerably across centres with the two residential centres offering fewer eligible participants (due to both the smaller size of the centres and a lower proportion of smokers). The two day-care centres were the most successful in terms of recruitment, together accounting for over two thirds of the total sample. Willingness to engage in the study at the centre in Northampton was particularly high and participants on the waiting list could not be recruited to the study due to criteria of recruiting across different sites. That we were able to recruit 80 participants across four centres in a five-month period despite the limited pool of participants at the residential centres is encouraging for a future main trial.

Retention is likely to be an issue when engaging with people accessing homeless centres as their circumstances can be unpredictable and quickly liable to change. The overall 24 week total retention rate was 59% which compares favourably with other smoking cessation studies in this population [6]. Retention was also much higher in the EC compared with the UC arm. This could reflect the nature of the intervention although the arms also differed on several important baseline characteristics with the UC group evidencing higher levels of long-standing illness, time spent in prison, substance dependence and anxiety which may militate against follow up attendance. However, there were also differences in retention rates between the 2 UC centres and the difference in retention between arms was driven largely by the low 24 week follow up rate at the Edinburgh centre where, despite our best efforts (text message reminders, incentives), many participants had moved on and were no longer accessing the homeless centre services.

Given the clinical and environmental factors specific to this population, it was recognised that EC may become lost, damaged, sold, or stolen. Encouragingly, reports of selling, exchanging or theft were infrequent. Although some reported breakages, most commonly through dropping the device which lead to broken tanks and electrical faults, the majority of participants still had and were still using either the EC that we provided or a different one. Critically, use continued after the end of the 4-week e-liquid supply period suggesting that our participants were willing to source their own e-liquid after the initial start-up costs are covered.

One concern that was not borne out, was that since 40% of smokers experiencing homelessness are dependent on other substances [61], there may be a temptation to use the EC for vaping illicit drugs. This is a trend that has been described in recent years by drug users in online forums, blogs and videos, though there is very little research on the topic. A recent study (a modest online survey using a convenience sample in the UK), reported that 39.5% of those using an EC had used it for recreational drug administration in their life time [35]. Most commonly reported was cannabis vaping; 18% of EC users reported lifetime use and 10.6% had used in the last 30 days. Although cannabis use was common in our sample, no participants reported adding cannabis (synthetic or otherwise) or any other illicit drug to the EC provided by the study during the 24 week study period.

Encouragingly, EC were well tolerated with high scores for self-reported positive effects e.g. 'pleasant' and 'tastes good'. Self-reported negative effects were rare; the highest scoring items on our VAS of negative effects were nausea and headache, both with an average severity score of less than 20%. Interestingly, these symptoms differ to those reported in other studies; using the same VAS for negative effects, we have previously found that throat irritation, mouth irritation and light-headedness are generally the most commonly experienced negative effects in smokers and EC users (although ratings are still low) whilst nervousness and headache are less typical [54, 62, 63].

As expected, our sample scored highly on measures of anxiety, depression, alcohol misuse and substance disorder at baseline. Encouragingly, illicit substance use (SDS) remained stable over time and GAD-7, PHQ-9 and AUDIT scores declined from baseline to follow up

suggesting that mental health and alcohol misuse do not get worse during a quit attempt and showed a slight improvement. This is consistent with previous research which has demonstrated that stopping smoking is associated with an improvement in mental health symptoms [34]. This is an important finding; in our recent systematic review, we report that professionals working within the homeless sector are often concerned about worsening mental illness or current substance use [6] and this has been a key barrier to both starting discussions around smoking and also implementing smoking cessation programmes. Health professionals should find the growing evidence that smoking cessation does not counter progress with other mental health symptoms in adults with complex needs reassuring. The potential benefits of cessation on mental wellbeing would be explored in a future trial.

E-cigarettes were well-received in the EC arm with the majority reporting that they still had, and were still using, the e-cigarette at the 24 week follow up. By 24 weeks, 31% had also made their own EC purchase which may explain why the number of people reporting use exceeded those who reported that they still had the EC we supplied. As has been reported in other feasibility studies of EC [64], and despite our cluster design, EC use was also commonly reported in the UC arm. We did not explore the extent of this use (e.g. regular use of their own device or just a few puffs of someone else's) and we did not explicitly ask UC participants not to use an EC. However, none of the UC participants received the EC intervention as delivered (i.e. a free EC starter kit offered at homeless centres). Although contamination across arms can be a problem for many RCTs, supported use of an EC is increasingly being incorporated into the 'usual care' offered by SSS and should not, therefore invalidate our findings. The pragmatic question that will be explored in a main trial will be whether the provision of a free EC starter kit, offered at a location already being accessed by people experiencing homelessness, can increase smoking cessation rates over usual care.

Our feasibility study results suggest that the EC intervention may have the potential to enhance smoking cessation rates in this population with a 6% sustained CO-verified abstinence rate at 24 weeks (intention to treat) compared to 0% in the UC group. Although cessation rates were low, if similar effects are found in a fully powered RCT, this could substantially contribute to reducing smoking-related inequalities given the high rates of smoking in this population. Notably, no other smoking cessation studies with people experiencing homelessness have reported continuous abstinence for 6 months [6]. However, our results compare favourably with the 24-hour point prevalence abstinence rate (4%) reported at 26 weeks by Segan et al. [13] and the 5.6% and 9.3% 7-day point prevalence at 26 weeks in a larger sample of 430 smokers experiencing homelessness who received either NRT or motivational interviewing + NRT respectively [12].

Completion rates of HRQoL and general healthcare and social services questionnaires suggest that the use of these measures was feasible in this population. The responses from the participants also indicated the services they most relied on and the direction of future refinement for data collection. Due to the situation this population find themselves in, certain services (e.g. home visit from GP) appear less relevant and others such as A & E and hospital stay, should be considered for more detailed investigation. Although the participants in the UC arm seemed to show a higher general healthcare and social services costs and worse quality of life, it should be noted that these results were based on observed cases and the retention in the UC arm was not as good as in the EC arm. It was also unclear if the baseline imbalance between arms was due to insufficient sample size or imbalance in other characteristics.

Nearly half of the total costs of intervention were contributed by training. This could be slightly underestimated as we did not include administration tasks done, in person, with e-mail or by phone, before and after actual training delivery. However, this should not add much to the mean costs after allocating evenly to all participants. Insufficient estimation of the

workload in participating centres meant that we were unable to capture what was happening in the UC arm in terms of their contacts with participants and how much efforts the keyworkers might have made. From self-reported SSS use, few participants in the UC arm sought SSS for quitting and they did not ask for help from GP services either after 4 weeks. While the EC arm did not seek SSS at all during the study period, they seemed to resort to GP services when they were no longer offered help by their keyworkers in the centre. This might indicate an attempt to keep on the efforts by some participants in the EC arm.

To our knowledge, this is the first study in the UK to engage with people accessing homeless services to support smoking cessation, and the first worldwide to explore the feasibility of supplying free EC starter kits. It is also the first to collect data on sustained abstinence for six months, the Russell gold standard usually adopted for smoking cessation studies [65]. Recruitment, retention, engagement and cessation rates compared favourably with previous studies. Our participants were open and willing to using EC and there were very few unintended consequences (e.g. device theft, adding illicit substances or deterioration in mental health). However, there were several limitations. Firstly, as a feasibility cluster randomised trial, we did not meet our original target recruitment rate. Nevertheless, this was a pragmatically chosen target based on information available to us at the time and there was further interest in participation if our schedule had allowed the researchers to remain on site for a longer period. Secondly, and relatedly, recruitment and retention differed across sites. This is an important finding, which, along with the results of our process evaluation (reported elsewhere) will assist us in carefully selecting sites (e.g. day-care only) in a future main trial. Thirdly, our arms differed on some important baseline characteristics. However, our intention was not to formally compare cessation rates between arms, but to inform the sample size and number of clusters required for a main trial. Fourthly, there was evidence of cross-contamination with EC use in the UC arm. This is a common issue in EC assisted smoking cessation research and one that will be carefully considered in the design of a future main trial. Finally, blinding was not possible for the measurement of outcomes which may have led to assessment bias or differential efforts on the part of researchers in contacting participants for follow up. Nevertheless, although follow up rates did differ between the arms, this appeared to be due to site differences rather than differences between treatment groups per se. In terms of generalisability, although our sample was relatively small, the data are drawn from three geographically distinct areas of GB, from a variety of different centres and the sample demographics match those of the wider homeless population (homeless.org).

## Conclusions

Our study demonstrated promising evidence of acceptability and efficacy of offering free EC starter kits to smokers accessing homelessness centres, a group that are among the most vulnerable and hard to reach in the UK. The findings of the study will be used to help inform the design of a main trial to definitively explore the efficacy of supplying EC starter kits to smokers accessing homeless services for smoking cessation in the UK.

## Supporting information

**S1 Checklist. TREND statement checklist.**
(PDF)

**S1 Table. Recruitment and retention by arm and centre.**
(DOCX)

**S2 Table. Recruitment and retention by arm and centre.**
(DOCX)

**S3 Table. Participants' smoking cessation help use at each time point, by arm.**
(DOCX)

**S4 Table. Number (of 10ml) bottles and types (flavour and strength) of e-liquid supplied by keyworkers each week.**
(DOCX)

**S1 Data.**
(SAV)

**S1 Protocol.**
(DOCX)

## Acknowledgments

We would like to thank the following for their support throughout the research process: Alicja Pytlik, Dr Catherine Kimber, and Emily Hussey.

We would especially like to thank the homeless charities involved: The Hope Centre, St Mungo's and The Salvation Army, and the support workers who provided exceptional levels of assistance and continual support.

We would also like to thank our Trial Steering Committee (TSC) members for their counsel and advice throughout the process.

## Author Contributions

**Conceptualization:** Lynne Dawkins, Linda Bauld, Allison Ford, Deborah Robson, Peter Hajek, Steve Parrott, Sharon Cox.

**Data curation:** Lynne Dawkins, Allison Ford, Steve Parrott, Allan Tyler, Isabelle Uny, Sharon Cox.

**Formal analysis:** Lynne Dawkins, Steve Parrott, Catherine Best, Jinshuo Li, Sharon Cox.

**Funding acquisition:** Lynne Dawkins, Linda Bauld, Allison Ford, Deborah Robson, Peter Hajek, Steve Parrott, Catherine Best, Sharon Cox.

**Investigation:** Lynne Dawkins, Allison Ford, Allan Tyler, Isabelle Uny, Sharon Cox.

**Methodology:** Lynne Dawkins, Linda Bauld, Allison Ford, Deborah Robson, Peter Hajek, Steve Parrott, Jinshuo Li, Sharon Cox.

**Project administration:** Lynne Dawkins, Allison Ford, Allan Tyler, Isabelle Uny, Sharon Cox.

**Supervision:** Lynne Dawkins, Linda Bauld, Peter Hajek, Sharon Cox.

**Writing – original draft:** Lynne Dawkins, Jinshuo Li, Sharon Cox.

**Writing – review & editing:** Lynne Dawkins, Linda Bauld, Allison Ford, Deborah Robson, Peter Hajek, Steve Parrott, Catherine Best, Allan Tyler, Isabelle Uny, Sharon Cox.

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
