## [Decision Letter · Decision Letter 0]

9 Sep 2020

PONE-D-20-19557

A cluster feasibility trial to explore the uptake and use of e-cigarettes versus usual care offered to smokers attending homeless centres in Great Britain

PLOS ONE

Dear Dr. Dawkins,

Thank you for submitting your manuscript to PLOS ONE. After careful consideration, we feel that it has merit but does not fully meet PLOS ONE’s publication criteria as it currently stands. Therefore, we invite you to submit a revised version of the manuscript that addresses the points raised during the review process and you will find below.

We look forward to receiving your revised manuscript.

Kind regards,

Christophe Leroyer

Academic Editor

PLOS ONE

Journal requirements;

2. Thank you for inncluding your competing interests statement;"I have read the journal's policy and the authors of this manuscript have the following competing interests:

PH received research funding from and provided consultancy to manufacturers of stop-smoking medications.

LD has provided consultancy for the pharmaceutical industry relating to the development of smoking cessation products.

SC, LB, AF, DR, SP, CB, AT, JL, IU, SP have no competing interests."

**Comments to the Author**

1. Is the manuscript technically sound, and do the data support the conclusions?

Reviewer #1: Yes

Reviewer #2: Yes

Reviewer #3: Yes

2. Has the statistical analysis been performed appropriately and rigorously? 

Reviewer #1: Yes

Reviewer #2: I Don't Know

Reviewer #3: Yes

3. Have the authors made all data underlying the findings in their manuscript fully available?

Reviewer #1: Yes

Reviewer #2: Yes

Reviewer #3: Yes

4. Is the manuscript presented in an intelligible fashion and written in standard English?

Reviewer #1: Yes

Reviewer #2: Yes

Reviewer #3: Yes

5. Review Comments to the Author

Reviewer #1: PONE-D-20-19557

Review of the paper A cluster feasibility trial to explore the uptake and use of e-cigarettes versus usual care offered to smokers attending homeless centers in UK

Global assessment

The proposed article is quite long and presents only a prospective limited randomized by cluster feasibility study which does not in itself allow definitive conclusions. On the principle it is the publisher's choice whether or not to accept this type of publication.

But this publication if of interest. This randomized trial concern:

1- tobacco dependence, the leading cause of preventable death. Tobacco related disease and mortality particularly affects vulnerable populations.

2- place of e-cigarette use for tobacco cessation is the object of strong discussions because of the lack of good data’s.

3- an innovative approach in precarious homeless people to quitting smoking through e-cigarettes is welcome

4- numerous very practical point for the conception of a large randomized trial are assessed in this paper.

This important article opens a door to the evaluation of e-cigarettes for nicotine replacement smoked in poor people who are heavy smokers and opens the door to improving health and reducing inequalities. The paper does not provide a definitive result but offers interesting results to launch a large essay that may provide the answer. This article provides an unprecedented series of information to understand the place of e-cigarettes for smokers in great social precariousness.

This paper is informative and clear and criticism are mainly of detail.

In detail

Length of introduction

The introduction (as the paper) is long and could be shortened.

Randomization page 8

The randomization process and result is not clear and had to be clarified.

It is clear that the randomization concern center and that all smokers in a same center received the same treatment.

• It seem that day centers and 24 h centers are associated in the random process.

• It appear than a center was changed after randomization.

• It appear than the smokers of a center had information of the result of the randomization in the center before inclusion.

• It seem than a center was designed to offer e-cigarette because of the proximity of the study center

I understand that the treatment was allocated by center, but the mode of distribution of treatment by center was not clearly identified (this is not a severe problem for a pilot study and lesson of this trial could be useful for a full size trial), but there is a need to better explain with coherence the allocation of treatment.

Page 9 recruitment of patients

It would be interesting to clarify what is said to smokers? Is the agreement for a smoking cessation need to be included? Is smoking reduction and option?

Is it clear to the patient that the trial is randomized?

Page 11 baseline(and follow-up) measure

The number of questionnaires included in the protocol addressed to homeless subjects is very high. Did the subjects complete the questionnaires on their own, were they completed with the help of a caregiver or by a signatory

NHS Treatement

Please precise what is an NHS treatment (Patch + oral nicotine) varenicline ?

Page 17 number of subjects in each arm

is there such a difference in the number of subjects in each arm (1/3 more subjects in the e-cigarette group than in the usual treatment group)?

Table 5 right column 1 to 6

I do not understand that in the right column we have frequencies at zero and percentages which are not zero for the first 6 lines. Please explain more or correct

Page 21 retention

I understand than at Week 4 only 25% of patients had reach SSS (no information on treatment) then in the EC arm 75% or 95% of patients use e-cigarette. But there is no or few data of the nicotine received daily in the two groups. Such information on daily dose and duration of nicotine received is important to assess in a complete trial the efficacy of treatment.

Page 29 cost of intervention

The duration of the visits seems short. The questionnaire completion time is often long in these homeless patients. 5 to 8 minutes for the follow-up visits seems very short, if not impossible, if this time includes the time for completing the questionnaire. If it is the health professional's time indicated: please specify the nature of these times

The cost needs to obtain one smoking cessation could be pertinent to be calculated in e-cig group. Not in UC group because no a single success.

Reviewer #2: Important note: This review pertains only to ‘statistical aspects’ of the study and so ‘clinical aspects’ [like medical importance, relevance of the study, ‘clinical significance and implication(s)’ of the whole study, etc.] are to be evaluated [should be assessed] separately/independently. Further please note that any ‘statistical review’ is generally done under the assumption that (such) study specific methodological [as well as execution] issues are perfectly taken care of by the investigator(s). This review is not an exception to that and so does not cover clinical aspects {however, seldom comments are made only if those issues are intimately / scientifically related & intermingle with ‘statistical aspects’ of the study}. Agreed that ‘statistical methods’ are used as just tools here, however, they are vital part of methodology [and so should be given due importance].

COMMENTS: Since this is a ‘feasibility trial’, sample size is not a big issue anyway. However, the objectives (given later on page-6) are not limited to (feasibility) use of e-cigarettes but comparison with usual care [offered to smokers attending homeless centres in Great Britain] is included. On page 15 [in Sample size section] it is said that “As this is a feasibility study, a formal power calculation based on detecting evidence for efficacy was not conducted”.

Objective three is “3. Assess the potential efficacy of supplying free e-cigarette starter kits to determine the required sample size for a main trial”. Intra class correlation coefficient (ICC) found in this trial will help to estimate the required sample size for a possible future definitive cluster RCT is understandable [however, need to mention this clearly]. But (because further you also said so) ‘how can the aim of the feasibility study was to calculate the required sample size?’ (please correct the sentence there).

Just a clarification – you said in ‘Methods section of Abstract’ that: “In this feasibility cluster trial, four homeless centres in Great Britain were pragmatically allocated to either a Usual Care (UC) or E-Cigarette (EC) arm” means the allocation was ‘non-random’ (in simple words). Is not that so? [dictionary meaning of ‘pragmatically’ is practically or rationally (therefore, what you said is perfectly alright) but we often describe allocation in terms of ‘random/non-random’]. Therefore, it is desirable to describe/classify allocation procedure in those terms.

Please note [for most data in this article through-out] that though measures/tools used are appropriate, most of them yield data in [at the most] ‘ordinal’ level of measurement [and not in ratio level of measurement as the score two times higher does not indicate presence of that parameter/phenomenon as double (for example, a Visual Analogue Scales {VAS} score or say ‘depression’ score)]. Then application of suitable non-parametric test(s) is/are indicated/advisable [even if distribution may be ‘Gaussian’ (i.e. normal)]. Note also that:

For ‘ordinal’ categorical data [as usual Chi-square test which otherwise is very versatile, the ‘ordering’ in categories is not recognized (example: instead of taking VAS scores are groups as No, Mild, Moderate, Severe, even if you take group sequence as Moderate, No, Mild, Severe, Chi-square value will be same), one variation called ‘Chi-square trend’ test is oftentimes recommended but] most recommended is RIDIT analysis [Bross IDJ. ‘How to use RIDIT analysis’. Biometrics, 1958; 14:18-38. There are many recent references, but they are on application results. This is (an original, old classic and) gives details on ‘how to use’ the technique].

In {last paragraph of} ‘Data analysis’ section, since it is stated that “For assessment of the relationship between other variables and outcomes, cases with missing data were excluded from the analysis“ I would like to know ‘how many cases were available?’. I doubt if ‘what is reported in Table 2’ [table titled as: Unintended consequences of E-cigarette use at 4, 12 and 24 weeks] are ‘consequences of E-cigarette use’. Findings of Table 4 [table titled as: Mental health and substance use at baseline and each follow up time point for the e-cigarette (EC) and usual care (UC) arms] are not properly interpreted, in my opinion {and displaying figures for ‘total’ is relevant, I guess}.

Note that when reported ICC, it is desirable to give the interpretation of ‘intraclass correlation coefficient’s point estimate is 0.0157?’ [though this Fleiss-Cuzick method is little known]. Nevertheless, there are few appreciable points, for example, inclusion of ‘Main RCT sample size calculation’ and discussion on ‘limitations’ [page 35].

Reviewer #3: This preliminary work provides important indications for a larger study to explore the feasibility of supplying free EC starter kits, targeting a vulnerable population, particularly at risk of smoking but also difficult to study rigorously in a clinical trial.

The questions are relevant and treated from a pragmatic angle.

For example, in practice, certain material challenges will have to be anticipated as best as possible such as, for example, the breakage of E-cigarettes...

The relatively encouraging results give an indication of the strengths and weaknesses of future studies, in particular the high proportion of those lost to follow-up during the study.

More generally, we must welcome the interest of a pragmatic approach that does not make guilt or infantilize towards a precarious and sensitive population.

6. PLOS authors have the option to publish the peer review history of their article (what does this mean?). If published, this will include your full peer review and any attached files.

Reviewer #1: **Yes: **Pr Bertrand Dautzenberg

Reviewer #2: No

Reviewer #3: No

---

## [Author Response · Author response to Decision Letter 0]

30 Sep 2020

1. We have reviewed the PLOS ONE style templates and changed our manuscript style in accordance with these templates.

2. We have updated our competing interests statement to include the required additional statement and this is included in our cover letter.

---

## [Editor Report · Decision Letter 1]

7 Oct 2020

A cluster feasibility trial to explore the uptake and use of e-cigarettes versus usual care offered to smokers attending homeless centres in Great Britain

PONE-D-20-19557R1

Dear Dr. Dawkins,

We’re pleased to inform you that your manuscript has been judged scientifically suitable for publication and will be formally accepted for publication once it meets all outstanding technical requirements.

Kind regards,

Christophe Leroyer

Academic Editor

PLOS ONE

---

## [Editor Report · Acceptance letter]

15 Oct 2020

PONE-D-20-19557R1 

A cluster feasibility trial to explore the uptake and use of e-cigarettes versus usual care offered to smokers attending homeless centres in Great Britain 

Dear Dr. Dawkins:

I'm pleased to inform you that your manuscript has been deemed suitable for publication in PLOS ONE. Congratulations! Your manuscript is now with our production department. 

Kind regards, 

on behalf of

Dr. Christophe Leroyer 

Academic Editor

PLOS ONE